# Coliphages as a Complementary Tool to Improve the Management of Urban Wastewater Treatments and Minimize Health Risks in Receiving Waters

**Juan Jofre [1,2], Francisco Lucena [1,3] and Anicet R. Blanch [1,3,*]** 

[1] Department of Genetics, Microbiology and Statistics, University of Barcelona, 08028 Catalonia, Spain; jjofre@ub.edu (J.J.); flucena@ub.edu (F.L.)

[2] Reial Acadèmia de Ciències i Arts de Barcelona, 08002 Catalonia, Spain

[3] Water Research Institute, University of Barcelona, Martí Franqueses 1, 08028 Catalonia, Spain

[*] Correspondence: ablanch@ub.edu

**Abstract:** Even in countries with extensive sanitation systems, outbreaks of waterborne infectious diseases are being reported. Current tendencies, such as the growing concentration of populations in large urban conurbations, climate change, aging of existing infrastructures, and emerging pathogens, indicate that the management of water resources will become increasingly challenging in the near future. In this context, there is an urgent need to control the fate of fecal microorganisms in wastewater to avoid the negative health consequences of releasing treated effluents into surface waters (rivers, lakes, etc.) or marine coastal water. On the other hand, the measurement of bacterial indicators yields insufficient information to gauge the human health risk associated with viral infections. It would therefore seem advisable to include a viral indicator—for example, somatic coliphages—to monitor the functioning of wastewater treatments. As indicated in the studies reviewed herein, the concentrations of somatic coliphages in raw sewage remain consistently high throughout the year worldwide, as occurs with bacterial indicators. The removal process for bacterial indicators and coliphages in traditional sewage treatments is similar, the concentrations in secondary effluents remaining sufficiently high for enumeration, without the need for cumbersome and costly concentration procedures. Additionally, according to the available data on indicator behavior, which is still limited for sewers but abundant for surface waters, coliphages persist longer than bacterial indicators once outside the gut. Based on these data, coliphages can be recommended as indicators to assess the efficiency of wastewater management procedures with the aim of minimizing the health impact of urban wastewater release in surface waters.

**Keywords:** fecal indicator; somatic coliphages; sewage treatment; water safety; surface water; marine coastal water

## 1. Introduction

To guarantee access to water and sanitation for all is goal number six of the 2030 Agenda of United Nations for Sustainable Development [1]. Sanitation, defined by the WHO as the provision of facilities and services for the safe disposal of human urine and feces, is still a pending problem in terms of controlling the impact of human residues on health. Although sanitation also includes the safe treatment of animal waste, given that human and animal feces share many microbes, including pathogens, this review is focused on the management of human fecal waste.

The importance of sanitation lies in the fact that waterborne pathogens are still one of the major public health concerns worldwide [2]. The global burden of disease in 2015 due to unsafe water resources has been estimated as 1.2 million deaths and 71.7 million disability-adjusted life years (DALYs), including 1.1 million deaths and 61.1 million DALYs from diarrheal diseases [3]. Although the problem affects mainly low- and medium-income

countries [4,5], water-related infections are not negligible in high-income nations such as the USA [6,7], Australia [8], and in Europe [9–13], where differences between eastern and western countries have been observed. The vast majority of waterborne pathogens are transmitted by the fecal–oral route and can re-infect humans through water used for drinking, recreation (bathing), irrigation (contaminating food), and shellfish farming (contaminating food). The aim of sanitation is to ensure the absence or minimize the presence of waterborne pathogens found in fecal remains in all these water resources.

Sanitation requires waste to undergo some sort of management. "On-site" sanitation services, which include septic tanks and dry toilets (pit latrines, composting dry toilets, urine-diverting dry toilets, etc.), if not managed correctly, contribute to the contamination of water sources by filtration to groundwater or through soil surface run-off to surface water bodies. "Off-site" sanitation involves the transport of wastewater through underground sewers. Sanitary sewers only carry wastewater generated in houses (black and grey waters) and small industries, whereas combined sewers, which are the majority, include additional water run-off (rain) from city streets and parks. Both kinds of wastewaters are referred to as municipal wastewater or sewage.

Habitually, raw wastewater and wastewater treatment plant (WWTP) effluents are discharged into water bodies (rivers, lakes, and seas) or soil. Otherwise, WWTP effluents are processed further to obtain reclaimed waters, which have a range of applications. The amount of treated urban wastewater is highly variable, being over 80% in high-income countries [14].

As mentioned, rivers, lakes, and seas are widely used as receiving waters for raw wastewater and WWTP effluents. The direct inflow of untreated or only partially treated wastewater often severely impairs the microbial quality of rivers and seawater. Even when good sanitation systems for urban wastewater are in place, spill-offs caused by failures across the service chain and overflow due to rain events result in the discharge of untreated fecal waste in the urban environment [15]. Consequently, even in regions with state-of-the-art wastewater treatment, such as Europe [16,17] and the USA [18], high levels of microbial fecal pollution, including pathogens, are found in surface and groundwater bodies. The association of this fecal contamination with waterborne infectious disease outbreaks is well documented [19–23].

Considering all the above, the aim of this review is to emphasize that the improvement in the microbial quality of treated wastewater needs further verification and that viral indicators, such as coliphages, should be included in wastewater management to monitor the treatment performance.

## 2. Worsening Prospects in the Near Future

A number of predictions for the coming years signal the need for reinforcing sanitation, including in wealthy countries with modern wastewater treatment facilities. The concentration of populations in large urban conurbations, climate change, aging of existing infrastructures, and emerging pathogens will increase the complexity of urban wastewater management.

More than half of the world population currently lives in urban agglomerations, a proportion expected to increase to almost seventy per cent in 2050, which will no doubt create severe sanitary challenges [24,25].

Foreseen climate changes, including a higher incidence of intense rainfall that produces water run-off and overflows in off-site sanitation services, are expected to increase the failure rate of sanitation systems even in developed countries [26–29]. Moreover, water scarcity due to climate change and the pressures arising from dense urban populations [30] will increase the need for water reclamation and reuse.

Due to aging and inadequate asset management, the wastewater collection infrastructures of many cities around the globe are in a state of rapid decline, resulting in the leakage of untreated sewage, with negative impacts on human and environmental health [31].

As recently evidenced by the emergence of SARS-CoV-2 [32], the appearance of new pathogens in the coming years cannot be ruled out. The emergence of a fecal–oral-transmitted pathogen, due to effective sanitation procedures and drinking water management, can be more easily controlled than a pathogen transmitted by air [33]. Nevertheless, a lack of immunity in the population will aggravate the effects of such outbreaks. An important cause of the emergence and re-emergence of infectious diseases is the growing resistance to anti-infective drugs. Fecal pollution is a major factor responsible for the abundance of antibiotic resistance genes in anthropogenically impacted environments [34].

Consequently, there is consensus that the achievement of "sanitation for all" will require a combination of different approaches involving various scales of technologies and services [25,35] and that even high-income countries need to take fecal microbial contamination into greater account in sanitation decision-making [36]. In light of the aforementioned facts, in 2020, the European Union decided to revise the Urban Wastewater Treatment Directive (Document Ares (2020)3769112) to protect both the environment and human health [37].

## 3. Pathogens and Fecal Indicators in Municipal Sewage

Assessment of sanitation processes and the contribution of raw or treated sewage to surface and underground water pollution first requires the effective monitoring of pathogens and microbes indicative of fecal contamination in sewage and treated waters, as well as their fate once released into the environment.

Sewage may contain microorganisms from several origins, but the immense majority derive from the microbiota in human feces [38], with minor input from the gut microbiota of animals living in sewer systems, such as rats and cockroaches. When combined with surface run-off water, urban sewage can contain microbiota from other sources, such as soil, pet feces, and environmental waters, but the human microbiota is still predominant [38].

Feces represent by far the greatest input of microbes into sewage: for a healthy individual producing an average of 100–200 g wet weight of feces per day, it contains an estimated $1.0 \times 10^{13}$–$2.0 \times 10^{13}$ bacteria [39]. Considering that the average daily contribution to sewage per person in high-income countries is 150–400 L of water, a liter of sewage will contain concentrations of $2.5 \times 10^{10}$ to $1.0 \times 10^{11}$ bacteria. Slightly higher concentrations of virus-like particles, ranging from $10^{11}$ to $10^{13}$ per liter, have been detected in raw sewage [40,41]. Genomic studies indicate that most of these viral particles correspond to bacteriophages [41].

The majority of human intestinal microbiota are anaerobic bacteria, such as *Bacteroides* and *Bifidobacterium*, many of them still uncultivable by current methods [39]. An initial qPCR-determined estimate of the concentration of *Bacteroides*, the dominant genus of bacteria in feces and raw sewage [42], supports the aforementioned number of total bacteria in the colon content. As detailed below, the contributing concentrations of bacteria used as indicators of fecal contamination, such as *E. coli*, enterococci, and sulfite-reducing clostridia, are several (3 or 4) log10 units lower.

Many pathogens, particularly viruses, as recently evidenced with SARS-CoV-2 [43], can be found in high concentrations in the urine and feces of infected individuals and hence are detected in sewage [44]. However, only those whose transmission via the fecal-oral route has been unequivocally established are considered worthy of attention from the sanitation and public health point of view. The number of pathogens, even in epidemic situations, is several log10 units lower than the number of bacteria and bacteriophages in the microbiota, the numbers depending on the sanitary status of the population, geographic region [45], and the season of the year [46]. Data on the numbers of fecal–orally transmitted pathogens reported in the scientific literature are available in previous reviews [38]. In theory, pathogen detection would seem to be an ideal option for managing sanitation and determining the microbiological quality of waters contaminated by sewage. However, such an approach is still neither practical nor feasible in routine testing due to geographic and temporal variations in prevalence, difficulties in detecting infectious pathogens, and the

uncertain ratios between infectious and non-infectious units determined by nucleic acid amplification, which vary in different settings and conditions [47].

The great majority of microbes of fecal microbiota, including pathogens and traditional fecal indicators, do not replicate outside the gut. Only a few genera of Proteobacteria, such as *Aeromonas* and *Pseudomonas*, replicate in sewers [48], but pathogens and indicators can survive transportation in sewer systems, wastewater treatments, and in nature. Survival occurs at different rates, resulting in variability in microbe proportions or relative concentrations as they are distanced in space and time from the polluting source.

## 4. Bacterial and Viral Fecal Indicators in Raw Sewage

For more than 100 years, fecal indicator bacteria (FIB), which are nonpathogenic bacteria of the intestinal microbiota, have been employed to assess both water quality and the efficiency of water treatments and management, and they are now included in guidelines and regulations all over the world. FIB are a diverse group of taxa whose selective detection and enumeration are made feasible by their phenotypic traits. They include total coliforms, thermostable coliforms (also reported as fecal coliforms), *E. coli*, enterococci (also reported as fecal streptococci or intestinal enterococci), and spores of sulfite-reducing clostridia [49,50]. Presence/absence and quantitative (colony forming units, CFU) culture-based methods standardized by regulatory agencies, as well as equivalent accredited methods developed by private companies as user-friendly kits, are available worldwide [50,51]. The resistance of FIB to treatments and their persistence in the environment are similar to those of bacterial pathogens [51], but their value as surrogate indicators of viruses and parasites has been questioned [52–54].

Efforts have been made over the last few decades to find fecal indicators that more closely mirror the behavior of viruses and parasites. Bacteriophages that infect enteric bacteria have been proposed as indicators of fecal pollution and/or viruses and are increasingly being included in water quality guidelines [55,56]. Feasible and cost-effective presence/absence and quantitative (plaque-forming units, PFU) methods standardized by regulatory agencies are available [57–60]. Moreover, fast and user-friendly methods that can be adapted for ready-to-use kits are being developed [61]. Both standardized [59] and fast methods are easily adaptable to 100 mL of water [62], thus avoiding the need to concentrate phages from volumes of up to 100 mL water samples.

Helminth eggs are also used as a parasite indicator for the management of wastewaters, mostly in low-income countries, where these parasites are still quite prevalent, with values ranging from <1 to $10^3$ per liter [49]. In contrast, in high-income countries, they are virtually absent, even in raw sewage.

Concentrations (CFUs and PFUs) of fecal indicators reported worldwide for 100 mL of incoming raw sewage at WWTPs are found in the following ranges: fecal coliforms/*E. coli* $10^6$–$10^8$; enterococci $10^5$–$10^7$; spores of sulfite-reducing clostridia $10^4$–$10^6$; somatic coliphages $5 \times 10^5$–$10^7$ and F-specific coliphages $10^5$–$10^6$ [49,63–68]. Concentrations of indicators in sewage collected in a given site vary according to various factors such as the fecal contribution to the sewage, occurrence of rain, and the time of day of sampling. Nevertheless, the relative proportions among the different indicators tend to remain constant.

Besides their concentrations, other features of fecal indicators in raw sewage, both bacterial and viral, are worthy of mention. Firstly, their concentrations in a given sewage collecting site show no seasonality [69,70], and secondly, their relative concentrations do not display geographical differences [67,71].

However, overflows in combined sewers (more rarely in sanitary sewers) due to heavy rainfall or snowmelt are responsible for a very high percentage of the fecal microbial load of the receiving waters, even when the overflows are modest in volume [72,73]. On the other hand, many sewer systems have significant accumulations of in-pipe deposits known as silt. Acting as a stockpile of pollutants, silt may exacerbate the detrimental impact of both combined and sanitary sewer overflows [74]. Field evidence indicates that 90% of the pollution load discharged from storm sewage overflows may be derived from silt

erosion [74]. In rural areas, zoonotic pathogens in surface run-off can also constitute a health risk [75], but this subject is not dealt with in the current review.

Though data are scarce, the proportions of bacterial and viral indicators in silt [76], combined sewage overflows [77] and urban coastal areas affected by sewage overflows [73,78] differ from those of raw sewage. As the relative concentration of coliphages is usually higher [77,78], albeit not always [73], they have interesting potential as additional indicators for the assessment of microbial fecal contamination in wastewater.

## 5. Removal of Pathogens and Indicators by Typical Sewage Treatment Plants

Wastewater treatment aims to produce an effluent that will do as little harm as possible to humans and nature when discharged to the surrounding environment, and cause minimal pollution compared to untreated wastewater. Acceptable levels of impurity will depend on whether the treated water is going to be reused or on the location of its disposal (surface water, groundwater, bathing or recreational zones, marine coastal water, etc.).

Most of the wastewater treatments currently used worldwide, including in member states of the European Union, where the procedures have to conform to the Urban Wastewater Treatment Directive, have been designed to remove particles and chemicals (mainly N and P). However, they also remove fecal microbes, both pathogens and indicators, which are mostly retained in sludges [79].

Commonly employed treatments comprise primary sedimentations plus one of the following: flocculation-aided sedimentation, activated sludge digestion, activated sludge digestion plus precipitation, and to a lesser extent, up-flow anaerobic sludge blanket processes and trickling filters. In all of them, microorganism die-off seems to play a minor role in the count reduction of pathogens and indicators, which accumulate in the resulting sludges that are subsequently treated.

In well-operated plants, the numbers of bacterial indicators, coliphages and pathogens undergo a similar decline. Reported reductions in the concentrations of naturally occurring infectious pathogens range from 0.3 to 3.0 log10 units, that is, from 50% to 99.9 %, depending on the treatment [80–87]. Thus, secondary effluents are still a source of pathogens, but the amounts vary depending on the season, epidemiological status of the population and the number of people served by the treatment facility.

Both bacterial and coliphage indicators are removed in ranges similar to pathogens, that is, from 0.3 to 3.0 log10 units, depending on the treatment [56,71,86,88–91]. Consequently, the ratios between bacterial and coliphage indicator concentrations, and between both types of indicators and naturally occurring pathogens in secondary effluents remain similar to those found in raw sewage. Coliphages and the most frequently used bacterial indicators are found in secondary effluents in numbers that can be detected without concentration using available procedures. Thus, the most frequent somatic coliphage values in secondary effluents range from $10^3$ to $10^5$ PFU per 100 mL [71,81,87,92].

As indicated earlier, secondary effluents are mostly discharged into surface waters when they are ecologically compatible with the surrounding environment and not intended for reuse after water reclamation treatment. However, some sensitive receiving water bodies, such as those used for bathing and shellfish collecting and farming, may require the effluents to undergo further processing prior to discharge, in which case chemical disinfection is common practice. According to most reports, such additional disinfection has a greater inactivation effect on bacterial than on bacteriophage indicators [93–95]. Of course, these observations do not refer to water reclamation and reuse, essential practices in the future to ensure a water supply for all, but which fall outside the scope of this review due to their large scale.

## 6. Coliphages in Wastewater-Receiving Surface Waters

As indicated previously, rivers, lakes, estuaries and seas commonly receive raw wastewater and WWTP effluents. Even in regions with state-of-the-art wastewater treatment, such as Europe [16,17] and the USA [18], high levels of microbial fecal pollution,

and hence coliphages, occur in surface water bodies. The coliphage densities in a given site of contaminated surface water are determined by the distance from outfalls, effluent volumes, the degree of dilution, sedimentation and inactivation of fecal microorganisms by natural stressors.

Intestinal microbes are excreted as aggregates, a fraction of which are found associated with particles in sewage [96]. On the other hand, in most natural conditions and environments, including WWTPs, coliphages, as viruses do, tend to adsorb to surfaces of solid particles [97,98], although attachment is variable due to environmental factors and the heterogeneity of different bacteriophage groups [98]. This behavior greatly affects the removal of coliphages from surface waters, as suspended solids facilitate their sedimentation. Moreover, viruses and bacteriophages adsorbed to surfaces tend to be less sensitive to anthropogenic and natural stressors and survive longer than when suspended in water [99,100]. Accordingly, coliphage concentrations detected in sediments outnumber by several orders of magnitude those in overlaying waters, both marine [101–103] and fresh [104–106]. The same applies for epilithic biofilms [106]. Increased river flow caused, for example, by storm events can re-suspend the sediments and detach phages from solids, thus reincorporating the coliphages into the water column [73,107].

Inactivation of coliphages in surface waters and sediments depends on different factors, both abiotic and biotic. The former include temperature, exposure to sunlight, the presence of natural photosensitizers and mineral and organic matter in the water [55,56,108–117]. Biotic factors such as predation and degradation caused by enzymes released by autochthonous microorganisms seem to play a minor role [109,118]. Although the results of some studies are ambiguous, the great majority of reports allow some general conclusions to be drawn. Coliphage numbers decline significantly faster when temperatures, salinities and sunlight exposure are higher. Most inactivation experiments report that coliphages mimic the abatement of viruses better than FIB, which generally decay faster. According to these observations, it can be predicted that the proportions of these groups of microorganisms change with the aging of polluted water.

A significant amount of information on coliphages and their relationship with FIB and pathogens has been collected in the last 30 years. The concentrations of coliphages in surface waters and their correlation with FIB and pathogens depends on several factors: firstly, the source of the coliphages, which are discharged into surface waters in treated or untreated urban wastewater and surface run-off, mostly of animal origin [60]; secondly, the level of inactivation, which depends on the distance from outfalls, the degree of dilution, sedimentation, and the age of the contamination; and finally, the diversity of methods used for detection and enumeration [56,109,119]. Table 1 summarizes the data obtained from various studies performed in a wide range of situations and sites. The reported concentrations of somatic coliphages are very diverse, because they correspond to areas with different contributions of fecal contamination, types of water, climate and distance from the pollution source. The studies also differ in the indicators and viruses they target and the methods applied. However, some general trends can be observed regarding somatic coliphages and FIB (*E. coli* /fecal coliforms), these parameters being reported in most of the studies. Numbers of coliphages and FIB are usually greater in freshwater than in seawater sites. The ratio between the numbers of *E.coli*/fecal coliforms and somatic coliphages is similar in wastewaters at freshwater sites, and both indicators are with high concentrations. This ratio diminishes in freshwater sites with lower concentrations of fecal contaminants, seawater, sites with aged fecal contamination and in dry periods. Data on infectious human viruses and other FIB are insufficient to make meaningful comparisons, though there is some evidence that compared to traditional indicators, coliphage densities are more strongly associated with viral pathogens.

**Table 1.** Concentrations of *E. coli*/fecal coliforms and somatic coliphages in surface waters. [a] Values of indicator bacteria and somatic coliphages are expressed as intervals or geometric means. In brackets percentages of positive samples, [b] values in MPN or CFU detected by methods according to national regulations, [c] ISO 10705–2 [57], [d] USEPA Method 1602 [59], [e] standard methods for the examination of water and wastewater [120].

| Samples | Somatic Coliphages Method | Number of Samples | Geographical Location | *E. coli* CFU/100 mL [a,b] | Somatic Coliphages PFU/100 mL [a] | Reference |
|---|---|---|---|---|---|---|
| Fresh water (river) | ISO [c] | 392 | Spain, France, Colombia, Argentina | $5.0 \times 10^3$ (100) | $6.2 \times 10^3$ (100) | [67] |
| Coastal and brackish water | USEPA [d] (strain C3000) | 12 | USA | $>4.0 \times 10^2$ (100) | 0.5. to $3.3 \times 10^2$ (100) | [121] |
| Freshwater (river) | ISO | 25 | South Africa | $1.1 \times 10^2$–$3.9 \times 10^4$ | $1.0 \times 10^2$–$7.7 \times 10^3$ | [122] |
| Freshwater (river) | ISO | 90 | Great Britain | $3.5 \times 10^3$ | $7.0 \times 10^3$ | [123] |
| Coastal water | APHA [e] | 20 | Malaysia | $1.5 \times 10^2$–$2 \times 10^4$ | 4-35 | [124] |
| Sea water | APHA | 61 | Brazil | $<1$–$8.4 \times 103$ (58) | $<1$–$3.4 \times 10^3$ (32) | [125] |
| Sea water | ISO | 806 | Spain | 30.1 (95) | 32.8 (72.6) | [126] |
| Fresh and sea water | ISO | 139 | Nine European countries | $1.0 \times 10^2$ (90) | $1.7 \times 10^2$ (92) | [65] |
| Freshwater (river) | ISO | 96 | France | $2.5 \times 10^2$ (100) | $3.0 \times 10^3$ (100) | [127] |
| Fresh and marine | ISO | 290 | Nine European countries | $3.0 \times 10^2$ (85) | $1.1 \times 10^2$ (72.5) | [128] |
| Fresh water (lake) | USEPA | 581 | USA | $2.0 \times 103$ (100) | $2.0 \times 10^2$ (96.4) | [129] |
| Estuarine water (lake) | USEPA | 222 | USA | 77 (100) | 30 (93.7) | [130] |
| Fresh water (river) | ISO | 23 | Japan | $10$–$3.2 \times 10^4$ (100) | $30$–$1.2 \times 10^3$ (100) | [73] |

The available data provide useful insights into the relationships between coliphages and FIB in surface waters and their potential significance, which should help in decision-making in the management of surface water quality to protect human health. What seems clear is that coliphages provide complementary information to that afforded by FIB. The identification of risk-based thresholds for coliphages from different hazards (treated wastewater or animal feces) or from mixed contamination of diverse sources and ages is an important subject for future research.

**7. Conclusions**

Although the treatment of wastewater before its release into the environment is a general practice in high-income countries, outbreaks of waterborne infectious diseases are still relatively frequent. Such health risks are expected to worsen in the future, due to the pressures arising from the growing concentration of populations in large urban conurbations, climate change, the aging of existing infrastructures and emerging pathogens. To meet this challenge, it is recommendable that wastewater management focuses more attention on the fate of pathogens.

At present, if effluents from municipal wastewater treatment plants are not treated further for water reclamation, they are usually discharged into the environment. Bacterial and coliphage indicators of water quality exhibit different degrees of resistance to water reclamation treatments and to environmental persistence [42,71]. On the other hand, the determination of either fecal coliforms or enterococci in water bodies does not provide sufficient information about the associated human health risk, especially regarding viral infections [56,131]. The concentrations of somatic coliphages in raw sewage remain consistently high throughout the year worldwide, as occurs with bacterial indicators. Moreover, the removal of bacterial indicators and coliphages in traditional sewage treatments is similar. Somatic coliphage concentrations in secondary effluents remain high enough for enumeration, without the need for cumbersome and costly concentration procedures. Additionally, coliphages persist longer than bacterial indicators once outside the gut, according to the available data on indicator behavior, which are still limited for sewers but abundant for surface waters.

Consequently, coliphages are being introduced in regulations for water reclamation [37,132] and bathing water quality [109]. Considering the current status quo, it seems judicious to include coliphage testing in the management of wastewater treatment plants, especially as coliphages can be enumerated faster than bacterial indicators [61,62,133], allowing decisions to be taken within one working day.

**Author Contributions:** Conceptualization, J.J., F.L. and A.R.B.; writing—original draft preparation, J.J.; writing—review and editing, F.L. and A.R.B. All authors have read and agreed to the published version of the manuscript.

**Funding:** This research received no external funding.

**Institutional Review Board Statement:** Not applicable.

**Informed Consent Statement:** Not applicable.

**Data Availability Statement:** This study did not report any data.

**Acknowledgments:** This work has been supported by the Spanish Government, Ministerio de Innovació y Ciencia (AGL2016-75536-P) and the Generalitat de Catalunya (2017SGR00170).

**Conflicts of Interest:** The authors declare no conflict of interest.

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
