# Peer review of "Coliphages as a Complementary Tool to Improve the Management of Urban Wastewater Treatments and Minimize Health Risks in Receiving Waters"

_water, doi:10.3390/w13081110_

Round 1
Reviewer 1 Report
The concept that phage indexicality is superior to bacterial indexicality is well known, but it is clearly discussed in this paper and is worth reading.
Tabel 1 L-3, 102 -> 10 power2
Author Response
Dear Reviewer:
We thank you for your time in reviewing the manuscript, for your comments, and for your suggestions. We have fixed the error you indicated.
Tabel 1 L-3, 102 -> 10 power2
Done
Yours,
A.R. Blanch
Reviewer 2 Report
This is a well-written and structured document.
The paper by Jofre and coauthors deals with the problem of monitoring the safety of water in terms of presence of pathogenic microorganisms.
In this framework, fecal indicator bacteria have been largely used to assess both water quality and the efficiency of processes such as disinfection. This indicator works well to indicate the presence of pathogens but it shows evident limitations in assessing the efficiency of disinfection processes.
In this paper the authors present a review of cases where colifages are used as a complementary tool to monitor the safety of water in a natural environment or after a treatment.
The paper is complete and well written and it can be published after minor revision.
The abstract is too generic should better highlight the results found by the literature review. The authors should highlight the aim of the paper and the degree of innovation at the end of the introduction section.
Line 99, the number and the year of the Directive should be added
Line 121, a short description of the method (qPCR) should be included
The language and the sentence structure can be improved.
Author Response
Dear Reviewer:
We thank you for taking the time to review the manuscript, for your comments, and for your suggestions that have helped us make the manuscript better.
Our answers to your comments and suggestions:
The abstract is too generic should better highlight the results found by the literature review. The authors should highlight the aim of the paper and the degree of innovation at the end of the introduction section.
We have revised the abstract as it suggests to us, highlighting the results and conclusions available in the literature. Modifications to the abstract are marked in yellow.
Line 99, the number and the year of the Directive should be added
The data and number of Document Ares has been indicated as suggested (marked in yellow)
Line 121, a short description of the method (qPCR) should be included
The description of the methods is outside the scope and approach of the manuscript. Including here a description of a specific qPCR would not be in coordination with the rest of the indications of other methods that are made throughout the manuscript. In addition, this qPCR technique is perfectly referenced so that any reader can go to the original source that describes it and that is perfectly cited. In any case, the purpose of the manuscript-review is not focussed to go into describing techniques.
The language and the sentence structure can be improved.
The text has been grammatically and linguistically corrected by a native English professional. Our manuscripts are always reviewed by professional foreign language services.
Yours,
A.R. Blanch
Reviewer 3 Report
Subject presented in this review article is rather important from the environmental and sanitation point of view. Problem concerns many countries all over the world. References which are the base of this paper (153 items) are up–to-date and relevant to the subject of paper.
Some comments:
- Title accurately describe the content of paper.
- In the abstract and in the end part of Introductions should by clearly defined the aim of paper.
- Conclusions should reinforce the scientific goal.
- Repeated sentences: line 63-65, and line 245-247.
- In References some explanation is needed, for example items: 54, 49, 106 (not clear edition).
Author Response
Dear Reviewer:
We thank you for taking the time to review the manuscript, for your comments, and for your suggestions that have helped us make the manuscript better.
Our answers to your comments and suggestions:
- Title accurately describe the content of paper.
Thanks
- In the abstract and in the end part of Introductions should by clearly defined the aim of paper.
We have revised the abstract as it suggests to us, highlighting the results and conclusions available in the literature. Modifications to the abstract are marked in yellow. An additional paragraph has been added at the end of the introduction to detail the aim of the paper as suggested. Both modifications are marked in yellow.
- Conclusions should reinforce the scientific goal.
We have added an additional paragraph to the conclusions to reinforce the scientific achievements and in coordination with the new text added to the abstract as another reviewer has suggested.
- Repeated sentences: line 63-65, and line 245-247.
Thanks. We have eliminated repetition on lines 245-247
- In References some explanation is needed, for example items: 54, 49, 106 (not clear edition).
We have reviewed the 3 citations indicated and they have been updated according to these same publications recommending “how to be cited”. We have also reviewed the other references in the manuscript. Thanks for pointing us to these mistakes.
Yours sincerely,
A.R. Blanch